

# Bioinformatic analysis and experimental validation of hub autophagy-related genes as novel biomarkers for type 2 diabetes mellitus and Alzheimer's disease

Rui Zhang[1,*], Ruowei Wang[2,*], Shuna Zhai[2], Chunhong Shen[2], Yu An[3] and Quanri Liu[1]

[1] Department of Nutrition, Beijing Luhe Hospital, Capital Medical University, Beijing, China
[2] College of Nursing, Shandong Xiehe University, Jinan, Shandong, China
[3] Medical Research Center, Beijing Institute of Respiratory Medicine and Beijing Chao-Yang Hospital, Capital Medical University, Beijing, China
* These authors contributed equally to this work.

Corresponding authors
Yu An, anyu900222@126.com
Quanri Liu, liuquanri8488@163.com

## ABSTRACT

**Background & Objectives:** Alzheimer's disease (AD) and type 2 diabetes mellitus (T2DM) share considerable similarities in their proposed patho mechanisms. Autophagy, an intrinsic cellular process involved in the degradation of dysfunctional organelles and abnormal proteins, has been implicated in the pathogenesis of both AD and T2DM. This study aims to identify potential shared biomarkers related to autophagy in AD and T2DM by analyzing hub differentially expressed autophagy-related genes (DEARGs) and examining their potential functions.

**Methods:** Gene expression profiles for AD and T2DM were acquired from the Gene Expression Omnibus (GEO) database (training sets: GSE109887 for AD and GSE104674 for T2DM; validation sets: GSE122063 for AD and GSE64998 for T2DM). Autophagy-related genes (ARGs) were extracted from multiple databases. DEARGs were identified and integrated with module genes derived from weighted gene co-expression network analysis (WGCNA) to determine key shared ARGs. Then, the STRING database was used to construct a protein-protein interaction (PPI) network, from which hub genes were identified. These hub genes were validated using independent microarray datasets through differential expression analysis, and ROC curves were generated to assess their diagnostic value. Moreover, the expression of the hub genes was validated in brain tissues of T2DM mouse models using qRT-PCR.

**Results:** A total of 33 shared DEARGs were identified, among which 12 were designated as hub genes (*ANXA5, CCND1, MAP2K1, HSPB1, BNIP3, BAG3, YAP1, MET, FBXW7, CCL2, PFKFB3, CDKN1A*) in both AD and T2DM patients. Validation using other datasets confirmed that *ANXA5, BAG3,* and *CDKN1A* remained significantly upregulated, while *MET* remained downregulated in both AD and T2DM patients. Additionally, *PFKFB3* showed an inverse expression pattern between the two diseases. The diagnostic performance of these five hub genes was assessed using ROC curves, with all five exhibiting values of area under the curve (AUC) exceeding 0.7 for T2DM in both training and validation sets. However, only *MET* and *PFKFB3* demonstrated good diagnostic efficacy in AD patients. In animal models, qRT-PCR analysis revealed that the expression of *ANXA5, BAG3,* and *MET*
was consistent with the bioinformatics results. In contrast, the expression of *PFKFB3* and *CDKN1A* did not differ significantly between db/db model mice and db/m control mice.

**Conclusions:** Our integrated bioinformatics analyses, supported by preliminary experimental validations, identified several hub ARGs shared between AD and T2DM. Among these, *ANXA5*, *BAG3*, and *MET* exhibited consistent expression trends across datasets and experimental models, while *CDKN1A* and *PFKFB3* showed inconsistent expression patterns. These findings underscore the complexity of autophagy-related crosstalk in AD-T2DM comorbidity and highlight the need for further research to clarify their diagnostic and therapeutic potential.

# INTRODUCTION

The association between cognitive impairment and type 2 diabetes mellitus (T2DM) is increasingly recognized as a significant comorbidity and complication (*Szablewski, 2025*). Individuals with T2DM exhibit approximately a 50% higher risk of cognitive decline, which in turn elevates their likelihood of developing dementia, with Alzheimer's disease (AD) being the most prevalent cause (*Yu et al., 2025*). AD is a progressive neurodegenerative disorder associated with aging and is characterized by cognitive impairment, memory loss, and behavioral disturbances. It is the primary cause of dementia and poses a significant financial and public health burden (*Zhang et al., 2024*). T2DM multifactorial metabolic disorder involving insulin resistance, chronic inflammation, endoplasmic reticulum stress, impaired autophagy, and mitochondrial dysfunction (*Xourafa, Korbmacher & Roden, 2024*). Additionally, T2DM also affects brain function, with pathological features such as amyloid beta (Aβ) accumulation, tau hyperphosphorylation, neuroinflammation, dysregulated autophagy, and oxidative stress—mechanisms that are likewise implicated in the onset and progression of AD (*Ye et al., 2023*). Despite mounting evidence supporting a robust association between T2DM and AD, the specific molecules and mechanisms responsible for this association have yet to be fully elucidated.

Recent studies have explored the intrinsic connection between T2DM and AD, identifying shared genes and pathways through bioinformatics approaches (*Ye et al., 2023*; *Wang & Yang, 2024*; *Huang et al., 2022*). For example, *Ye et al. (2023)* successfully mapped overlapping transcriptional changes between AD and T2DM; however, their analysis did not specifically focus on autophagy-related genes (ARGs). Recent reviews have underscored the critical role of autophagy in the pathogenesis of T2DM and AD (*Yang et al., 2024*; *Hein et al., 2025*). Some mechanistic studies also revealed autophagy's multifaceted role in AD and T2DM, respectively. *Brown et al. (2022)* demonstrated that increased expression of REV-ERBα in pancreatic β-cells under diabetogenic stress impairs

autophagy and contributes to β-cell failure in T2DM. *Choi et al. (2023)* showed that inhibition of microglial autophagy results in microglial disengagement from amyloid plaques, suppression of disease-associated microglia, and exacerbation of neuropathology in AD. Given the pivotal role of autophagy in both diseases, investigating ARGs could provide deeper insights into their shared pathogenesis. In this context, *Caberlotto et al. (2019)* revealed autophagy as a centrally dysregulated pathway through a network-based analysis of transcriptomic data of postmortem human AD and T2DM brains, complemented by validation in an AD mouse model. However, previous studies have faced several limitations, including limited application of advanced bioinformatics analysis methods, such as weighted gene co-expression network analysis (WGCNA) and protein-protein interaction (PPI), to resolve tissue-specific co-expression modules and provide valuable information on shared biological processes. Additionally, previous studies have been limited by the lack of systematic integration of ARGs from multiple databases and the absence of experimental validation using T2DM comorbidity models. To address these gaps, the present study applies advanced bioinformatics techniques to comprehensively identify novel ARGs associated with both T2DM and AD. These findings are further validated in comorbid animal models, with a focus on autophagy-specific biomarkers that may inform the development of targeted therapies for T2DM and AD.

## METHODS AND MATERIALS

### Data collection and ARGs datasets

Datasets relevant to AD and T2DM were systematically curated from the Gene Expression Omnibus (GEO) database (https://www.ncbi.nlm.nih.gov/geo/) to align with the primary objective of this study, which focuses on these two distinct diseases. The inclusion criteria for dataset selection were as follows: (1) a minimum of 15 samples in total, (2) at least five individuals in both the patient and control groups, and (3) the availability of either raw data or processed gene expression profiles. Consequently, the following GEO accession datasets were utilized: (1) GSE109887 for AD (training set), consisting of mRNA expression profiles from brain tissue of 46 AD patients and 32 healthy controls; (2) GSE104674 for T2DM (training set): consisting of mRNA expression profiles of subcutaneous adipose tissue from 24 T2DM patients and 24 healthy controls; (3) GSE122063 for AD (validation set), consisting of mRNA expression profiles from brain tissue of 44 AD patients and 56 healthy controls; (4) GSE64998 for T2DM (validation set): mRNA expression profiles from liver tissue of 14 T2DM patients and seven healthy controls.

ARGs were obtained on October 28, 2021, from multiple databases, including HADb (http://www.autophagy.lu/), AUTOPHAGY DATABASE (http://autophagy.info), HAMdb (http://hamdb.scbdd.com/), and MsigDB (https://www.gsea-msigdb.org/gsea/msigdb/index.jsp). For MsigDB, 19 gene sets closely associated with autophagy were selected using "autophagy" as the keyword, encompassing a total of 557 genes.

Additionally, GeneCards (http://www.genecards.org/) was queried using the same keyword "autophagy", and genes with a relevance score of ≥1.0 were retained. This search yielded a total of 2,181 ARGs (Table S1).

## Differential gene expression analysis

Following data preparation for each disease, differential expression analysis of autophagy-related genes (DEARGs) was performed using the "limma" R package for the AD and control groups in the GSE109887 dataset, and the "edgeR" package for the T2DM and control groups in the GSE104674 dataset. In both analyses, genes were considered significantly differentially expressed if they met the thresholds of $|\log_2(\text{fold change})| > 0.3$ and a false discovery rate (FDR)-adjusted $P$-value < 0.05. The FDR threshold (<0.05) was applied to control the expected proportion of false positives among the identified DEARGs to ≤5%, a standard and stringent correction for multiple testing in transcriptomic studies. The $|\log_2 FC| > 0.3$ threshold (corresponding to approximately 1.23-fold linear changes) was selected to: (1) filter out technical noise and focus on biologically relevant changes; (2) adopt a relatively inclusive approach to detect potentially important genes with subtle expression differences, which is particularly relevant in complex comorbid conditions; and (3) account for the fact that modest expression changes (*e.g.*, 1.2–1.5 fold) in key regulatory genes can have substantial biological impact, especially in neurodegenerative contexts. Therefore, these thresholds were chosen to balance sensitivity and specificity, enabling the identification of subtle but consistent transcriptional changes associated with disease comorbidity, while minimizing false positives through FDR correction. Similar thresholds have been used in published transcriptomic studies of comorbid diseases (*Qi et al., 2023*). Differential expression results were visualized using volcano plots.

## Weighted gene co-expression network analysis (WGCNA)

To identify gene modules with similar co-expression patterns and to investigate hub genes within the network and their association with disease status (T2DM or AD), we performed weighted gene co-expression network analysis (WGCNA) separately on each dataset (GSE109887 for AD and GSE104674 for T2DM) to account for tissue-specific expression profiles. The soft-thresholding power (β) was determined using the scale-free topology criterion (scale-free $R^2 > 0.85$) to ensure that the resulting adjacency matrix approximated a scale-free network. The adjacency matrix was constructed by raising the absolute value of the Pearson correlation coefficient between gene pairs to the power β ($a_{ij} = |\text{cor}(i, j)|\beta$). A power of β = 4 satisfied this criterion for both datasets. The topological overlap matrix (TOM) was then derived from this adjacency matrix to measure network interconnectedness. A hierarchical clustering dendrogram was generated based on TOM-based dissimilarity (1-TOM). Modules were identified using the dynamic tree cutting algorithm, and those with highly correlated module eigengenes (height < 0.25) were merged. The correlation between each module eigengene and disease status (AD or T2DM *vs* control) was calculated using Pearson's correlation coefficient, and modules significantly associated ($P < 0.05$) with disease status were selected for further analysis.

## Shared gene identification

DEARGs from each disease dataset were intersected with genes from WGCNA modules significantly correlated with the respective disease status, resulting in a set of shared ARGs potentially involved in the pathogenesis of both AD and T2DM.

## Functional enrichment analysis of DEARGs

Functional enrichment analysis of the 33 shared genes for KEGG pathways and GO terms was performed using the R package "clusterProfiler". GO enrichment analysis was performed across three categories: biological process (BP), cellular component (CC), and molecular function (MF). Pathways or terms with an FDR-adjusted corrected $P$ value < 0.05 were considered statistically significant.

## PPI network construction and differential expression analysis of hub DEARGs

A PPI network was constructed to identify key hub ARGs shared between AD and T2DM. The shared DEARGs were input into the STRING database to generate the PPI network. Hub genes within this network were identified based on nod degree, which represents the number of direct connections a node (protein) has within the network. A higher degree values indicate greater potential functional significance. Genes with a degree >3 were defined as candidate hub genes. Among them, the top 12 genes exhibiting the highest degree values were ranked and designated as key hub genes. To investigate differences in their expression, the expression levels of these key hub DEARGs were compared across normal, AD, and T2DM groups, and the results were visualized using boxplots.

## External datasets validation and prediction accuracy assessment of hub genes

To validate the hub genes, we analyzed their expression in two independent datasets for AD and T2DM (GSE122063 and GSE64998) and the results were visualized using boxplots. The diagnostic performance of the hub genes was evaluated using ROC curve to differentiate AD/T2DM from normal controls and calculate the area under the curve (AUC) to evaluate their diagnostic value.

## Validation of hub genes in animal models

To investigate the expression of hub genes in the brains of T2DM models, 6-week-old male db/db mice ($n = 8$) and age-matched male littermate db/m controls ($n = 8$) were purchased from the SiPeiFu (Beijing) Biotechnology Co., Ltd. Upon arrival, the mice underwent a 1-week acclimatization period followed by 5 weeks of experimental feeding, resulting in a total age of 11 weeks at the study endpoint. The mice were housed under specific-pathogen-free (SPF) conditions with a 12 h/12 h light/dark cycle, controlled temperature ($20 \pm 2$ °C), and relative humidity (55%). Bedding was regularly replaced, and cages were routinely cleaned and disinfected. Four mice were housed in each cage to meet social needs, with environmental enrichment provided in the form of nesting materials and shelters. After 5 weeks of feeding, body weight and fasting blood glucose (FBG) levels were

measured to confirm hyperglycemia in db/db mice compared to controls. FBG levels were determined through tail vein sampling. In accordance with institutional guidelines, no analgesia was administered due to the minimally invasive nature of the procedure. At the end of the study, mice were euthanized by intraperitoneal injection of entobarbital sodium (50 mg/kg body weight) and cervical dislocation. Brains were immediately harvested, dissected on an ice-cooled board, snap-frozen in liquid nitrogen, and stored at −80 °C for qRT-PCR analysis. All animal procedures were approved by the Animal Ethics Committee of Capital Medical University (AEEI-2023-296) and performed according to the relevant ethical standards.

## Morris water maze test

The Morris water maze (MWM) test was conducted to evaluate spatial learning and memory abilities. Briefly, the MWM test was conducted in a circular black pool filled with water and a hidden platform. The pool was surrounded by black curtains to eliminate external visual cues and was divided into four quadrants: northeast, northwest, southeast, and southwest. During the acquisition phase, mice underwent four trials per day for 5 consecutive days. Each trial ended when the mouse located the hidden platform or after 60 s had elapsed. Escape latency (the time taken to locate the platform) and swim paths were recorded using an overhead camera. On the final day, a probe trial was conducted by removing the platform, and mice were allowed to swim for 60 s. The number of platform crossings and the time spent in the target quadrant were recorded using a video tracking system (JX business, Shanghai, China).

## Quantitative qRT-PCR analysis

After the MWM test, all mice were sacrificed by cervical dislocation. The brains were immediately harvested, dissected on an ice-cooled board, snap-frozen in liquid nitrogen, and stored at −80 °C for quantitative qRT-PCR analysis. Total RNA was extracted using TRIzol reagent (Invitrogen, Waltham, MA, USA), and its purity and concentration were evaluated by measuring absorbance at 260 and 280 nm using an ultramicro UV-visible spectrophotometer. Complementary DNA (cDNA) was synthesized from the RNA using the Hifair® III 1st Strand cDNA Synthesis SuperMix (Yeason, Shanghai, China), according to the manufacturer's instructions. The reverse transcription reaction was carried out at 55 °C for 15 min, followed by 85 °C for 5 s. Quantification of mRNA expression was performed through qRT-PCR using SYBR Green PCR kit (SAIPUBIO, Shenzhen, China) on a FQD-96A real-time PCR system (Bio-Rad, Hercules, CA, USA). All primer sequences were synthesized by Sangon Biotech (Shanghai, China). *GAPDH* was used as the internal reference gene for normalization. Relative gene expression levels were calculated using the $2^{-\Delta\Delta Ct}$ method. Primer sequences are shown in Table 1.

## Statistical analysis

Statistical analyses were carried out using SPSS 29.0 (SPSS, Inc., Chicago, IL, USA) and GraphPad Prism 9.0 (GraphPad Software, San Diego, CA, USA). Data are presented as

**Table 1 The detailed sequences of all primers used for RT-qPCR.**

| Genes | Forward primer | Reverse primer |
|---|---|---|
| GAPDH | CCTCGTCCCGTAGACAAAATG | TGAGGTCAATGAAGGGGTCGT |
| ANXA5 | CTGTTTGGCAGGGACCTTGTG | TAGGCATCGTAGAGTCGTGAGG |
| BAG3 | TCCATTCAGGTCACCCGTCA | CCTGGCTTACTTTCTGGTTTGTTT |
| MET | CAGACGCCTTGTACGAAGTGATG | ATGAACGTGGAGAAGATTGAGGA |
| PFKFB3 | CTCTTACAACTTCTTCCGCCCT | ACTGCAATCTGTCCACCTTCCTT |
| CDKN1A | TCCAATCCTGGTGATGTCCG | GTCAAAGTTCCACCGTTCTCG |

**Table 2 Summary of those four GEO datasets involving AD and T2DM patients.**

| Diseases | GEO series | Samples | Source types | Group |
|---|---|---|---|---|
| AD | GSE109887 | 46 patients and 32 controls | Brain | Training set |
| T2DM | GSE104674 | 24 patients and 24 controls | Subcutaneous adipose tissue | Training set |
| AD | GSE122063 | 44 patients and 56 controls | Brain | Test set |
| T2DM | GSE64998 | 14 patients and 7 controls | Liver | Test set |

**Note:**
    AD, Alzheimer's disease; T2DM, type 2 diabetes mellitus.

mean ± SD. Data were compared between two groups using the independent samples t-tests. Escape latency in the MWM test was analyzed using repeated-measures ANOVA. A two-tailed $P < 0.05$ was considered significant.

# RESULTS

## Identification of DEARGs

In this study, four GEO datasets (GSE109887, GSE104674, GSE122063, and GSE64998) were analyzed, and their information is summarized in Table 2. ARGs were compiled from multiple databases, including HADb, the Autophagy Database, HAMdb, MsigDB, and GeneCards, yielding a total of 2,181 ARGs. Differential expression analysis was performed using the thresholds of $|\log_2 FC \text{ (fold change)}| > 0.3$ and FDR-adjusted $P$ value $< 0.05$. Based on these criteria, 2,416 DEARGs were identified between brain tissue samples from AD patients and normal controls using the GSE109887 dataset (Fig. 1A). Similarly, 1,882 DEARGs were obtained in subcutaneous adipose tissue samples from T2DM patients and normal controls using the GSE104674 dataset (Fig. 1B).

## Construction of co-expression modules by WGCNA in AD and T2DM

WGCNA was performed on AD samples from the GSE109887 dataset to identify gene co-expression modules associated with AD status (Fig. 2). The soft-thresholding power was determined based on the criterion of achieving a scale-free topology with a fit index ($R^2$) > 0.85, resulting in an optimal power value of 4. With this threshold, a weighted co-expression network was constructed using a one-step approach (Fig. 2A). Most genes exhibited a low degree of connectivity, while a few genes displayed high connectivity, consistent with a scale-free network topology indicative of hub genes. The linear regression analysis yielded $R^2$ values of 0.86 and 0.84, with slopes of −1.08 and −1.52, respectively,

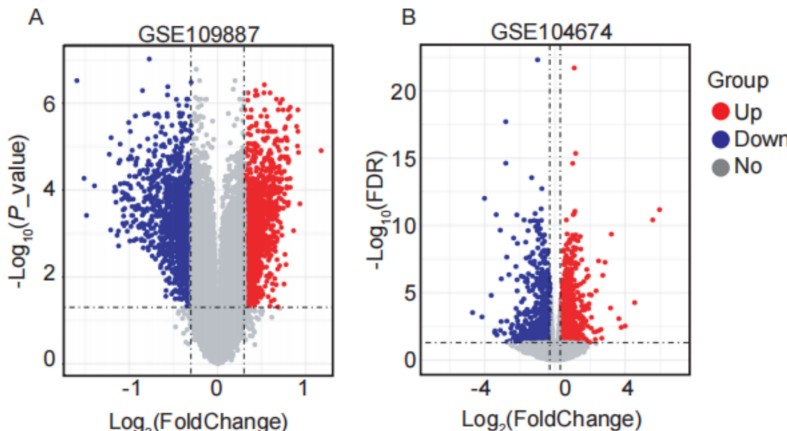

**Figure 1 Volcano plots of differentially expressed autophagy-related genes (DEARGs) in AD and T2DM training datasets.** (A) DEARGs between AD patients ($n = 46$) and controls ($n = 32$) in brain tissue (GSE109887). (B) DEARGs between T2DM patients ($n = 24$) and controls ($n = 24$) in subcutaneous adipose tissue (GSE104674). Red dots: upregulated genes ($\log_2$FC > 0.3, FDR-adjusted $P < 0.05$); blue dots: downregulated genes ($\log_2$FC < −0.3, FDR-adjusted $P < 0.05$); gray dots: non-significant genes.

further supporting the scale-free nature of the constructed network (Fig. 2B). Hierarchical clustering and module merging (height cut-off < 0.25) yielded five distinct gene modules (Fig. 2C). Subsequent correlation analysis between module eigengenes and AD clinical traits demonstrated that the blue, brown, and green modules were significantly associated with AD status ($P < 0.05$, Fig. 2D). Collectively, these three modules encompassed 800 genes, which were selected for downstream analysis as potential AD-related biomarkers. Figure 2 provides a comprehensive visualization of the WGCNA workflow and outcomes for AD.

In the T2DM analysis, WGCNA was applied to the GSE104674 dataset to identify co-expression networks associated with the pathogenesis of T2DM (Fig. 3). Consistent with the approach used for AD, the soft-thresholding power was determined based on the scale-free topology criterion ($R^2 > 0.85$), yielding an optimal power value of 4 (Figs. 3A and 3B). The network was constructed with this power, and hierarchical clustering with a height cut-off of <0.25 for module merging produced seven gene modules (Fig. 3C). Correlation analysis with T2DM-associated clinical traits revealed that five modules—blue, brown, yellow, green, and red—were significantly correlated with T2DM status ($P < 0.05$, Fig. 3D). These modules collectively contained 913 genes, which were considered potential T2DM-related markers. The detailed outcomes are depicted in Fig. 3, which captures the WGCNA process for T2DM.

## Identification of shared genes

To investigate the autophagy-related co-pathogenesis of AD and T2DM, we conducted an intersection analysis between the previously mentioned DEARGs and the genes identified by WGCNA. As shown in Fig. 4, a total of 33 genes (*ANXA5, BAG3, BNIP3, CCL2, CCND1, CDKN1A, COX5A, DNM3, FBXW7, GABARAPL1, GLS, GSTP1, HSPB1, HSPB8,*

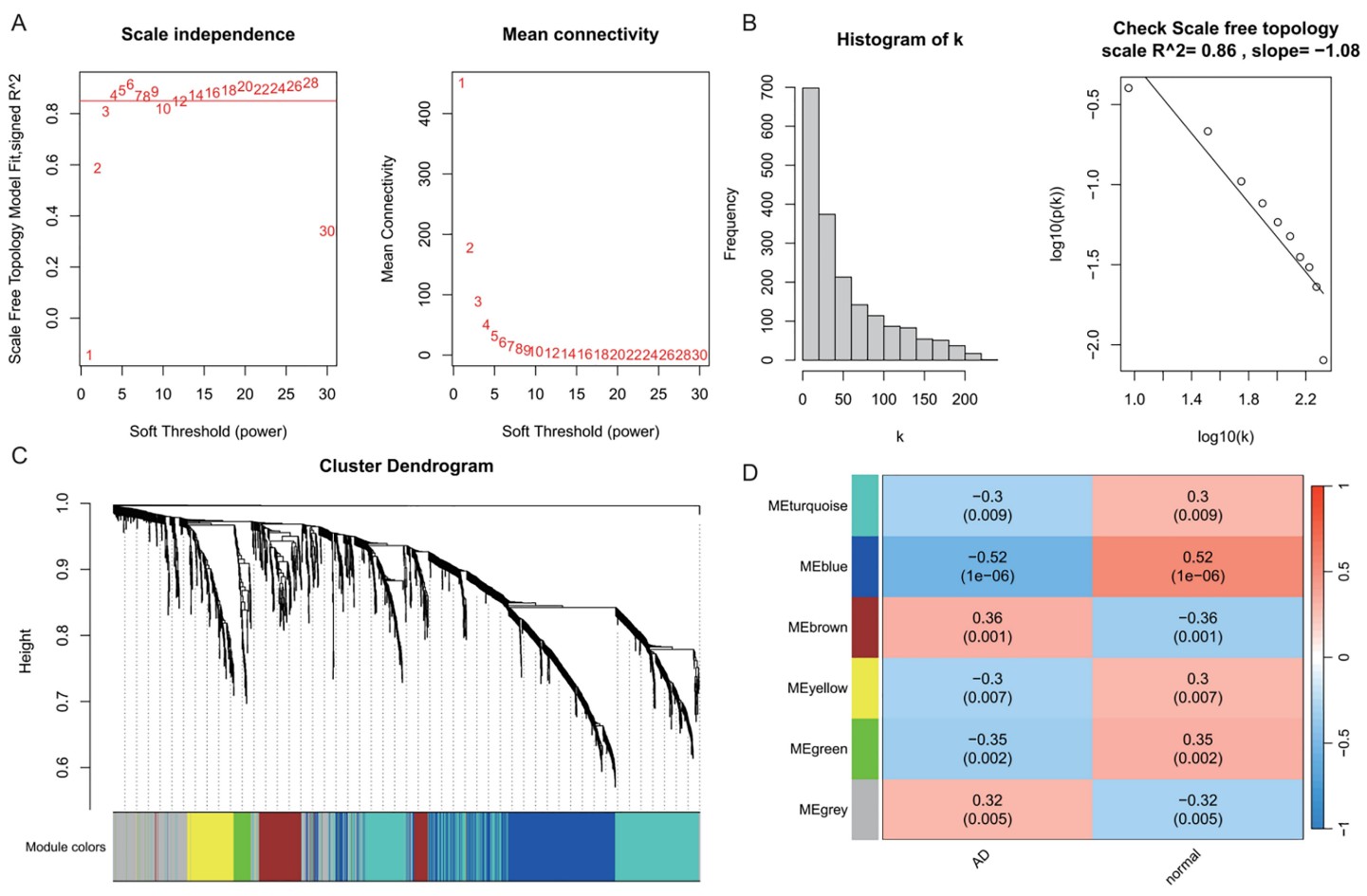

**Figure 2 Weighted gene co-expression network analysis (WGCNA) identifying disease-associated modules in Alzheimer's disease (AD).** (A) Soft threshold selection: displaying the scale-free topology fit index ($R^2$) across varying soft threshold powers. X-axis of scale independence represents soft thresholding power ($\beta$), Y-axis shows scale-free topology model fit ($R^2$). Mean connectivity across nodes decreases with increasing power values. As the power increases, the mean connectivity decreases, resulting in a sparser network that better meets the scale-free topology fitting requirement. The optimal power of 4 was selected where $R^2 > 0.85$, ensuring a scale-free network structure. Red line: soft threshold power ($\beta = 4$) (B) Connectivity distribution detection: illustrating the distribution of connectivity (number of connections per gene/node) within the network. Most genes exhibit low connectivity, while a few genes (hub genes) exhibit high connectivity, demonstrating a scale-free characteristic. A log-log plot of log (k) (logarithm of connectivity) vs log(p(k)) (logarithm of frequency) demonstrates that the linear fit yields an $R^2 = 0.86$ and a slope = −1.08, confirming the scale-free nature of the constructed network. (C) Clustering results: showing the hierarchical clustering results based on gene similarity. The X-axis represents genes, and the Y-axis represents the clustering height (distance). The color bars below the dendrogram denote the six modules identified using the dynamic tree cut algorithm. A merge cut height threshold of height < 0.25 was applied to merge similar modules into larger ones. Gray: genes not assigned to modules. (D) Module-trait correlation calculation: presenting the correlation (pearson correlation coefficient) and its statistical significance (*P*-value) between a module eigengene and a disease trait, with turquoise, blue, brown, yellow and green modules showing significant positive or negative correlations with AD status. Color scale: red (positive correlation), blue (negative correlation). Cell values: correlation coefficient (top) and *P*-value (bottom).

KIF5B, MAP1B, MAP2K1, MET, NUAK1, PFKFB3, PHGDH, PLK2, PRKCH, RAB6B, RPN2, S100A4, TBC1D9, TNS3, TSPO, TUBB2A, TXNIP, UCHL1, YAP1) were identified. We hypothesize that these genes may play critical roles in the autophagy-related pathogenesis of both AD and T2DM and may represent a set of shared molecular targets (Fig. 4).

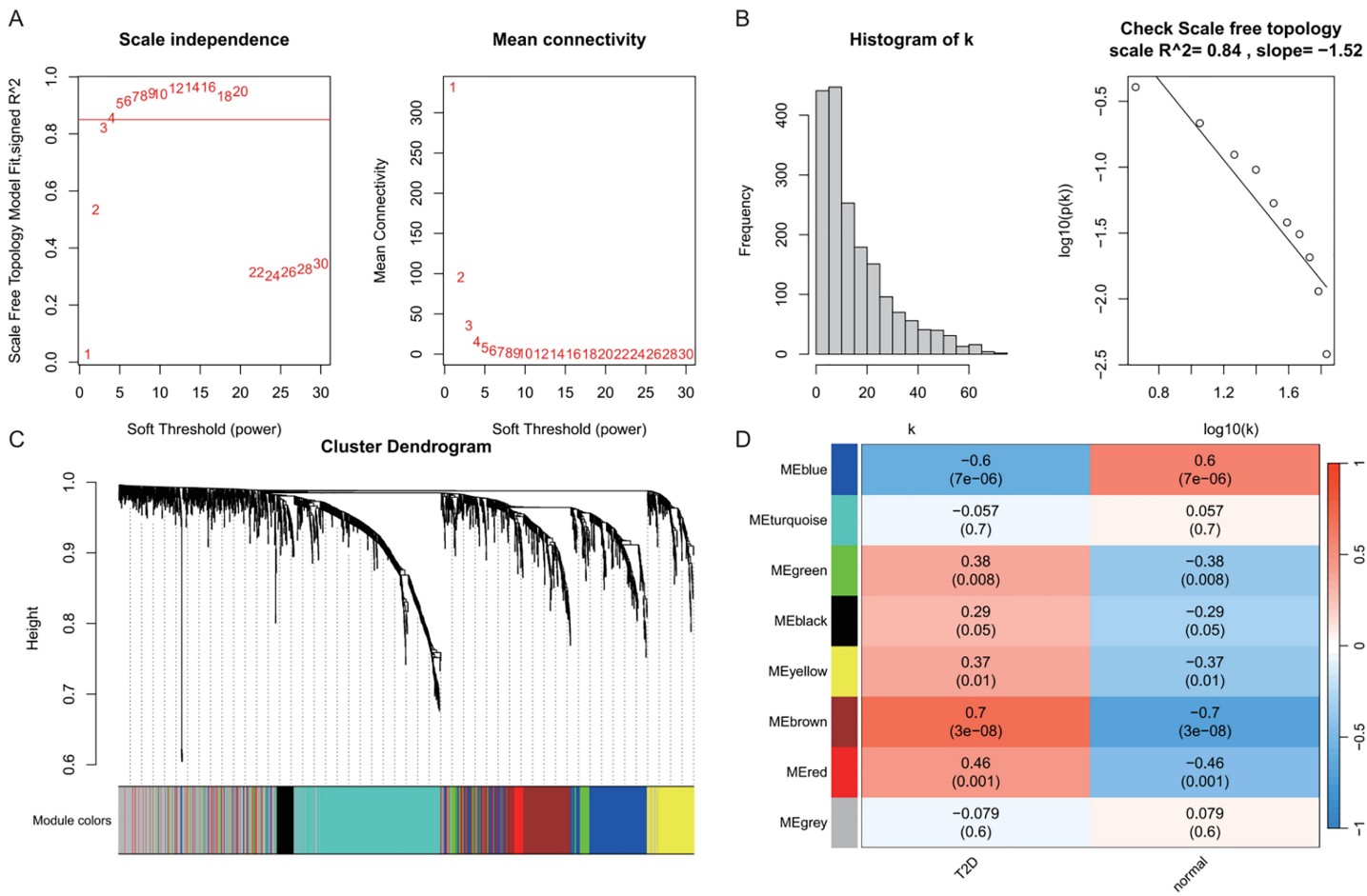

**Figure 3 Weighted gene co-expression network analysis (WGCNA) identifying disease-associated modules in T2DM.** (A) Soft threshold selection: displaying the scale-free topology fit index ($R^2$) across varying soft threshold powers. X-axis of scale independence represents soft thresholding power (β), Y-axis shows scale-free topology model fit ($R^2$). Mean connectivity across nodes decreases with increasing power values. As the power increases, the mean connectivity decreases, resulting in a sparser network that better meets the scale-free topology fitting requirement. The optimal power of 4 was selected where $R^2 > 0.85$, ensuring a scale-free network structure. Red line: soft threshold power (β = 4). (B) Connectivity distribution detection: illustrating the distribution of connectivity (number of connections per gene/node) within the network. Most genes exhibit low connectivity, while a few genes (hub genes) exhibit high connectivity, demonstrating a scale-free characteristic. A log-log plot of log(k) (logarithm of connectivity) *vs* log(p(k)) (logarithm of frequency) demonstrates that the linear fit yields an $R^2 = 0.86$ and a slope = −1.08, confirming the scale-free nature of the constructed network. (C) Clustering results: showing the hierarchical clustering results based on gene similarity. The X-axis represents genes, and the Y-axis represents the clustering height (distance). The color bars below the dendrogram denote the six modules identified using the dynamic tree cut algorithm. A merge cut height threshold of height < 0.25 was applied to merge similar modules into larger ones. Gray: genes not assigned to modules. (D) Module-trait correlation calculation: presenting the correlation (pearson correlation coefficient) and its statistical significance (*P*-value) between a module eigengene and a disease trait, with blue, brown, yellow, green, and red modules with significant associations to T2DM. Color scale: red (positive correlation), blue (negative correlation). Cell values: correlation coefficient (top) and *P*-value (bottom).

## Function annotation of shared genes

To investigate the biological roles of the 33 shared genes, GO and KEGG enrichment analyses were performed. As shown in Figs. 5A–5C, GO enrichment analysis revealed that the shared genes were primarily enriched in autophagy-related BP, including "autophagy," "process utilizing autophagic mechanism," and "regulation of autophagy" (Fig. 5A). In

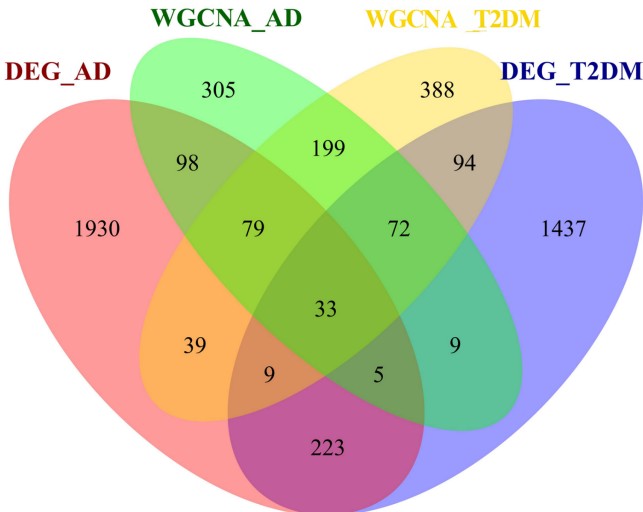

**Figure 4 Venn diagram of shared autophagy-related genes between AD and T2DM.** Intersection of DEARGs and WGCNA module genes identified 33 shared genes.

terms of CC, the genes were mainly associated with the "microtubule" and "neuron projection cytoplasm" (Fig. 5B). For MF, enrichment was observed in "kinase regulator activity" and "ubiquitin protein ligase binding" (Fig. 5C). KEGG pathway analysis further indicated that these genes were predominantly enriched in the FoxO signaling pathway and pathways related to various types of cancer (Fig. 5D).

## PPI network construction and identification of hub genes

The 33 shared genes between AD and T2DM were used to construct a PPI network *via* the STRING database. As indicated in Fig. 6A, the resulting network was visualized using Cytoscape. Based on node degree ranking, 12 genes (*ANXA5, CCND1, MAP2K1, BAG3, BNIP3, HSPB1, CCL2, FBXW7, MET, YAP1, CDKN1A* and *PFKFB3*) were identified as hub genes for further analysis (Fig. 6B). Moreover, the expression of these hub DEARGs were evaluated in the GSE109887 (AD) and GSE104674 (T2DM) datasets, comparing patient samples with controls (Figs. 7A and 7B). Seven hub genes, including *ANXA5, BAG3, CCL2, CCND1, CDKN1A, HSPB1*, and *YAP1*, were significantly upregulated in both the brain tissues of AD patients and the subcutaneous adipose tissues of T2DM patients. In contrast, *BNIP3* and *MET* were significantly downregulated in both conditions, showing consistent expression patterns across AD and T2DM. However, three hub genes exhibited divergent expression trends between the two diseases. Specifically, *PFKFB3* gene was significantly upregulated in the brain tissues of AD patients but markedly downregulated in the adipose tissues of T2DM patients. In contrast, *FBXW7* and *MAP2K1* were significantly downregulated in AD but upregulated in T2DM, demonstrating opposite expression patterns between the two conditions.

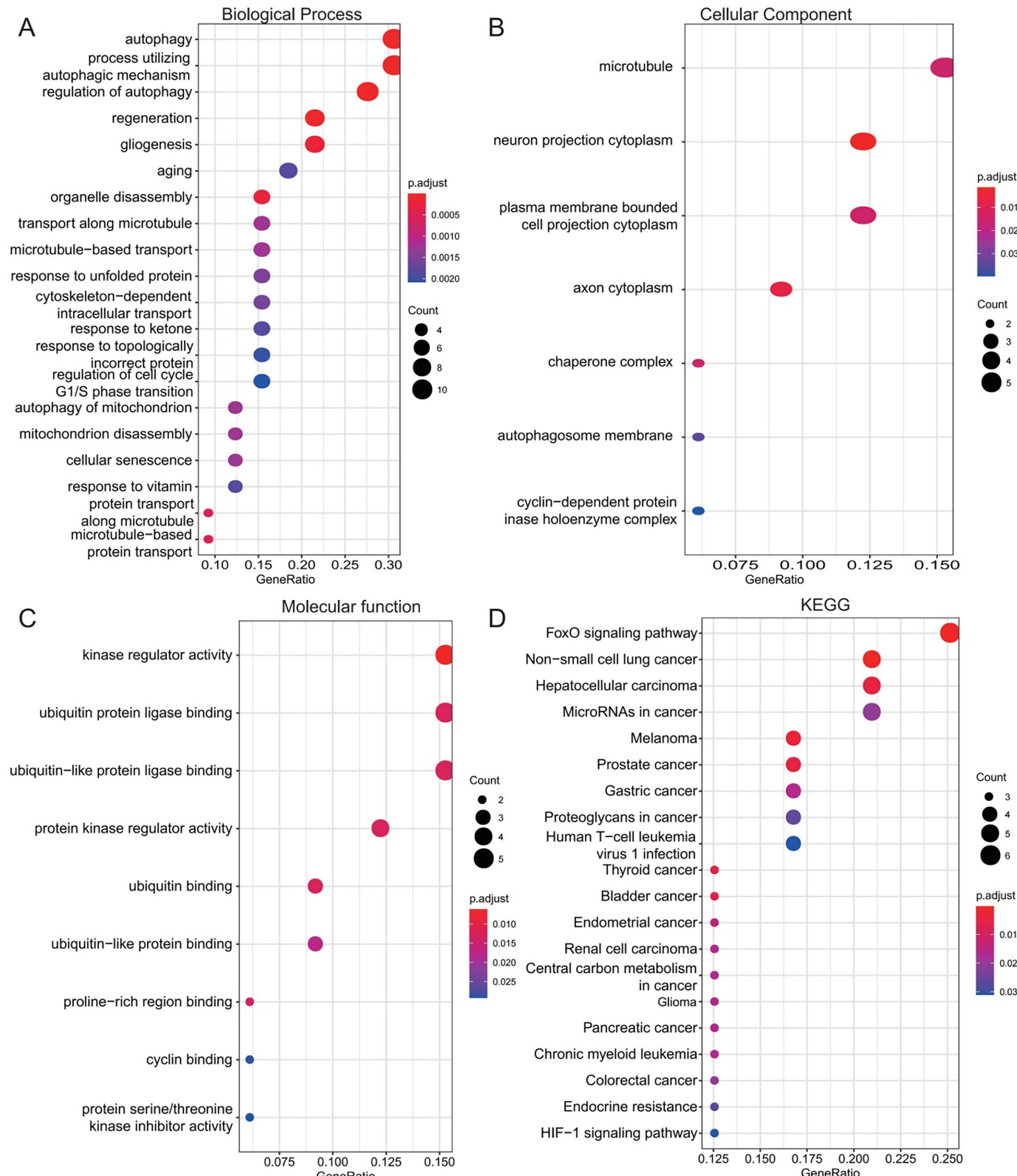

**Figure 5 Functional enrichment analysis of 33 shared autophagy-related genes.** The bubble plot of the BP (A), CC (B) and MF (C) category of GO enrichment and KEGG pathway enrichment analysis (D) for 33 shared genes between AD and T2DM identified by overlapping their DEARGs

**Figure 5 (continued)**
and WGCNA. The color intensity of nodes represents the adjusted *P*-value, and the node size indicates the number of genes. BP, biological processes; CC, cellular components; MF, molecular function; GO, gene ontology; KEGG, Kyoto Encyclopedia of Genes and Genomes.

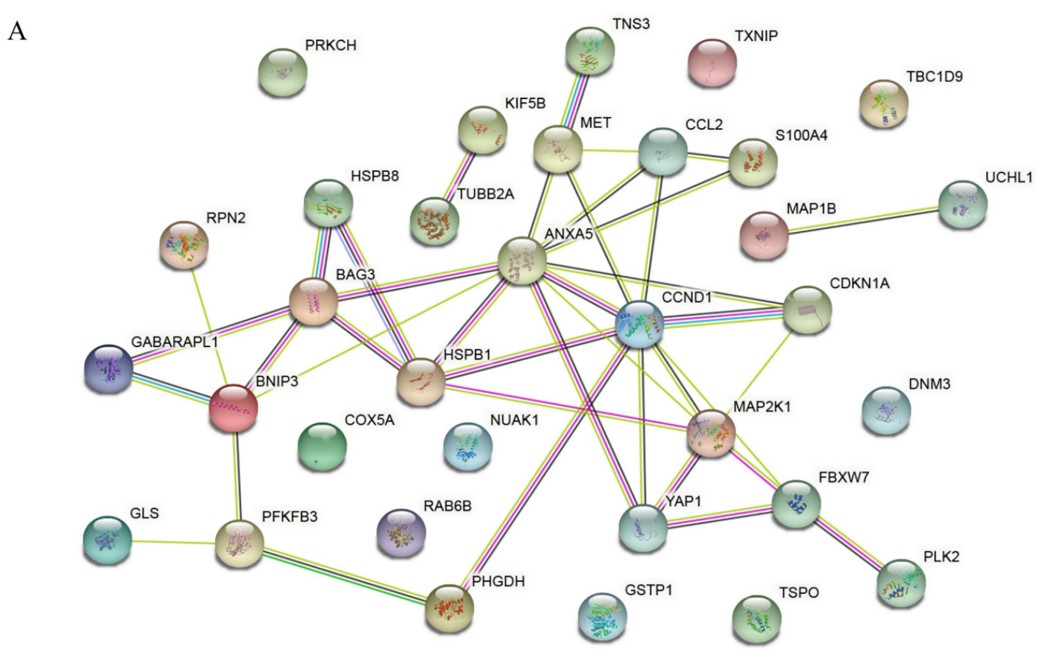

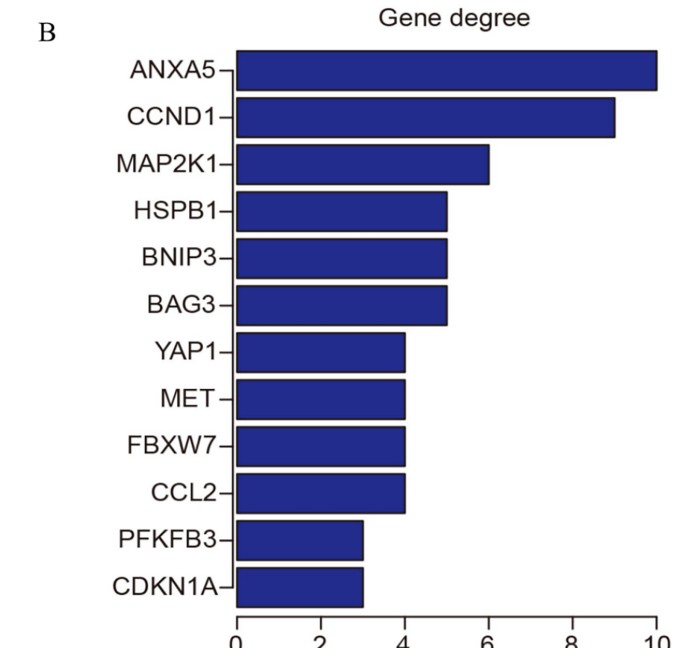

**Figure 6 Protein-protein interaction (PPI) network and hub gene identification.** (A) PPI network of 33 shared genes. Nodes represent the 33 shared autophagy-related genes; edges indicate functional or physical interactions. (B) Twelve hub genes filtered by degree (top 12 nodes).

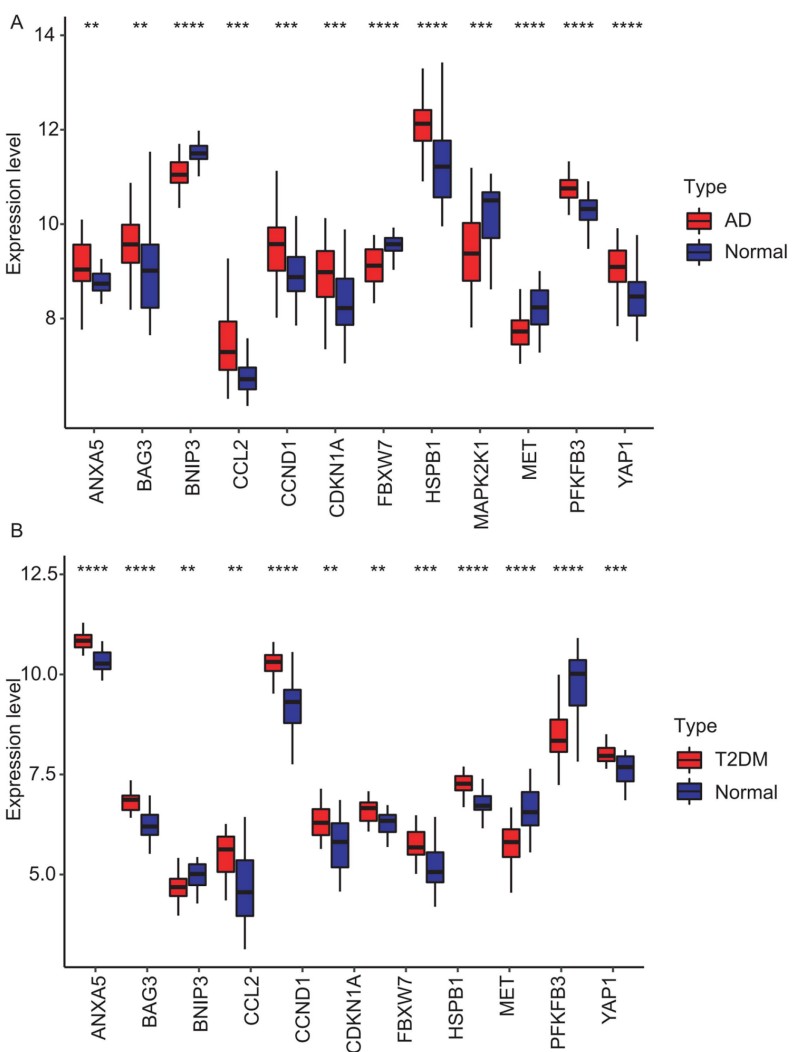

**Figure 7 Expression levels of 12 hub genes in AD and T2DM training datasets.** Expression differences between disease and control groups were assessed using the Wilcoxon test, with significance thresholds: \*\*$P < 0.01$, \*\*\*$P < 0.001$, \*\*\*\*$P < 0.0001$. Results are visualized as boxplots (median and quartiles). (A) AD dataset (GSE109887): all 12 hub genes showed significant alterations ($P < 0.05$) in AD brain tissue ($n = 46$) *vs* controls ($n = 32$). (B) T2DM dataset (GSE104674): all 12 hub genes showed significant alterations ($P < 0.05$ ) in T2DM adipose tissue ($n = 24$) *vs* controls ($n = 24$).

## External datasets validation and diagnostic efficacy assessment of hub genes

To validate the identified hub genes, we analyzed two independent datasets: GSE122063 (AD) and GSE64998 (T2DM). To analyze differential expression patterns of hub genes (Figs. 8A and 8B). Differential expression patterns of the 12 previously identified hub genes were examined. Expression levels were compared between brain tissues of AD patients and controls, as well as between subcutaneous adipose tissues of T2DM patients and controls. The analysis revealed that *ANXA5, BAG3,* and *CDKN1A* remained significantly upregulated, while *MET* remained downregulated in both AD and T2DM patients.

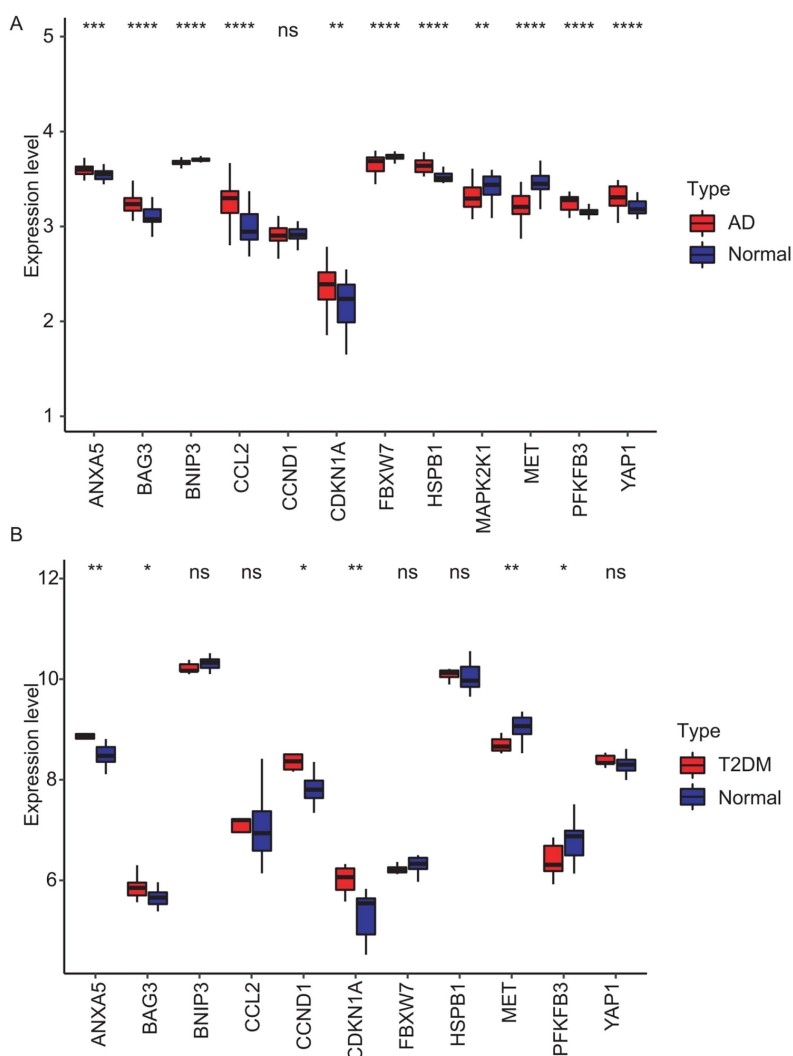

**Figure 8 Validation of hub gene expression in independent test datasets.** Expression differences between disease and control groups were assessed using the Wilcoxon test, with significance thresholds: *$P < 0.05$, **$P < 0.01$, ***$P < 0.001$, ****$P < 0.0001$. Results are visualized as boxplots (median and quartiles). (A) AD dataset (GSE122063): 11 hub genes remained significantly dysregulated in independent AD brain samples ($n = 44$) *vs* controls ($n = 56$). (B) T2DM dataset (GSE64998): Only six hub genes were validated in T2DM liver tissue ($n = 14$) *vs* controls ($n = 7$).

Conversely, *PFKFB3* exhibited an opposite expression pattern between the two. However, *BNIP3* and *FBXW7* were significantly downregulated, and *CCL2, HSPB1*, and *YAP1* were upregulated only in AD patients. In contrast, *CCND1* was significantly upregulated only in T2DM patients.

The diagnostic performance of five hub ARGs (*ANXA5, BAG3, CDKN1A, MET*, and *PFKFB3*), which demonstrated significant differential expression in both the training and validation sets, was evaluated using ROC curve analysis. Among these genes, *ANXA5, BAG3*, and *CDKN1A* were significantly upregulated in both AD and T2DM patients, while *MET* was significantly downregulated in both groups, suggesting that the expression of

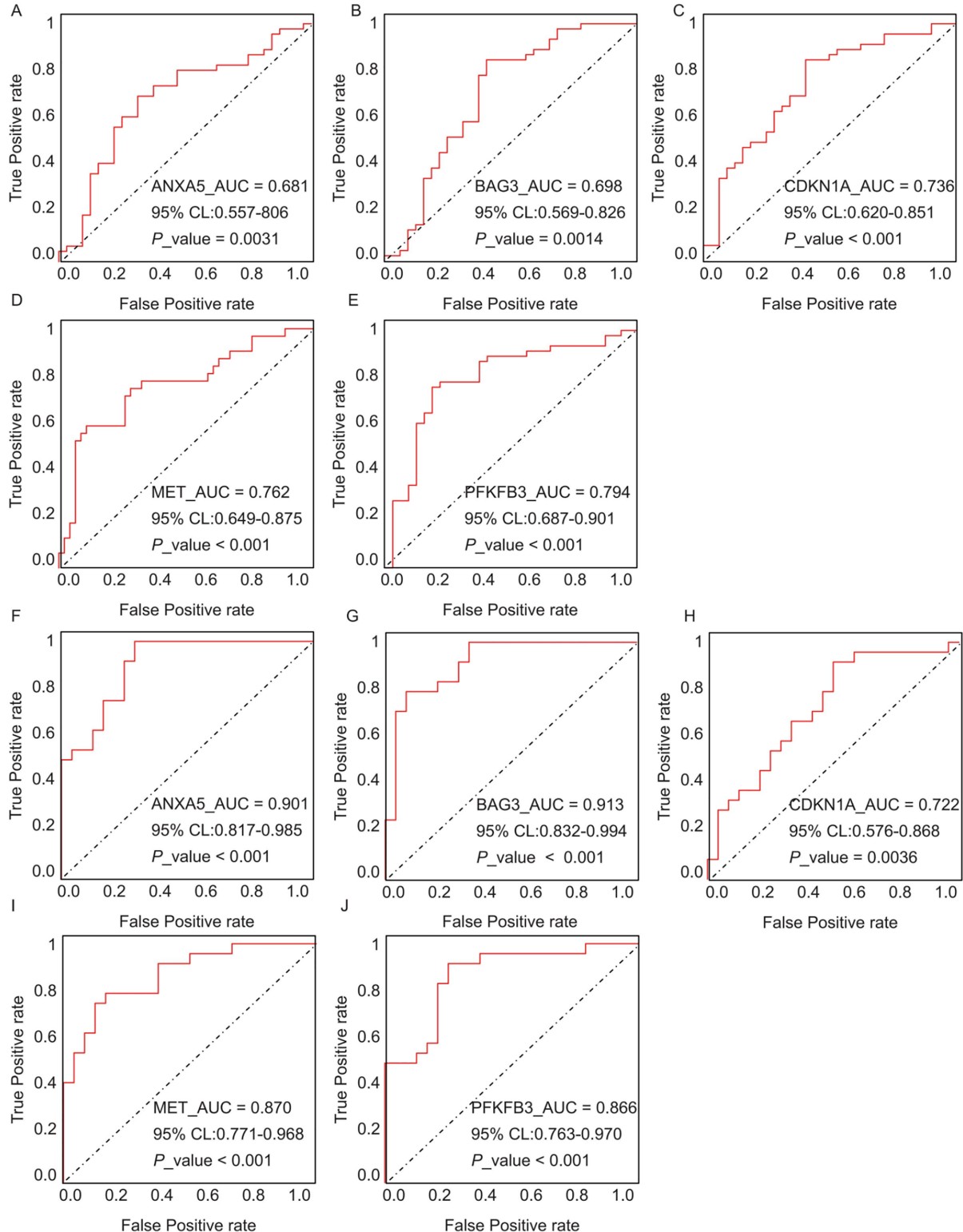

**Figure 9 ROC curves of the five hub genes (ANXA5, BAG3, CDKN1A, MET, and PFKFB3) derived from the training datasets (GSE109887 for AD and GSE104674 for T2DM), namely *ANXA5* for AD (A), *BAG3* for AD (B), *CDKN1A* for AD (C), *MET* for AD (D), *PFKFB3* for AD (E), *ANXA5* for T2DM (F), *BAG3* for T2DM (G), *CDKN1A* for T2DM (H), *MET* for T2DM (I), *PFKFB3* for T2DM (J).** Each curve illustrates the trade-off between sensitivity (true positive rate) and 1-specificity (false positive rate), with the AUC values, 95% confidence intervals (CIs), and P-values quantifying diagnostic accuracy.

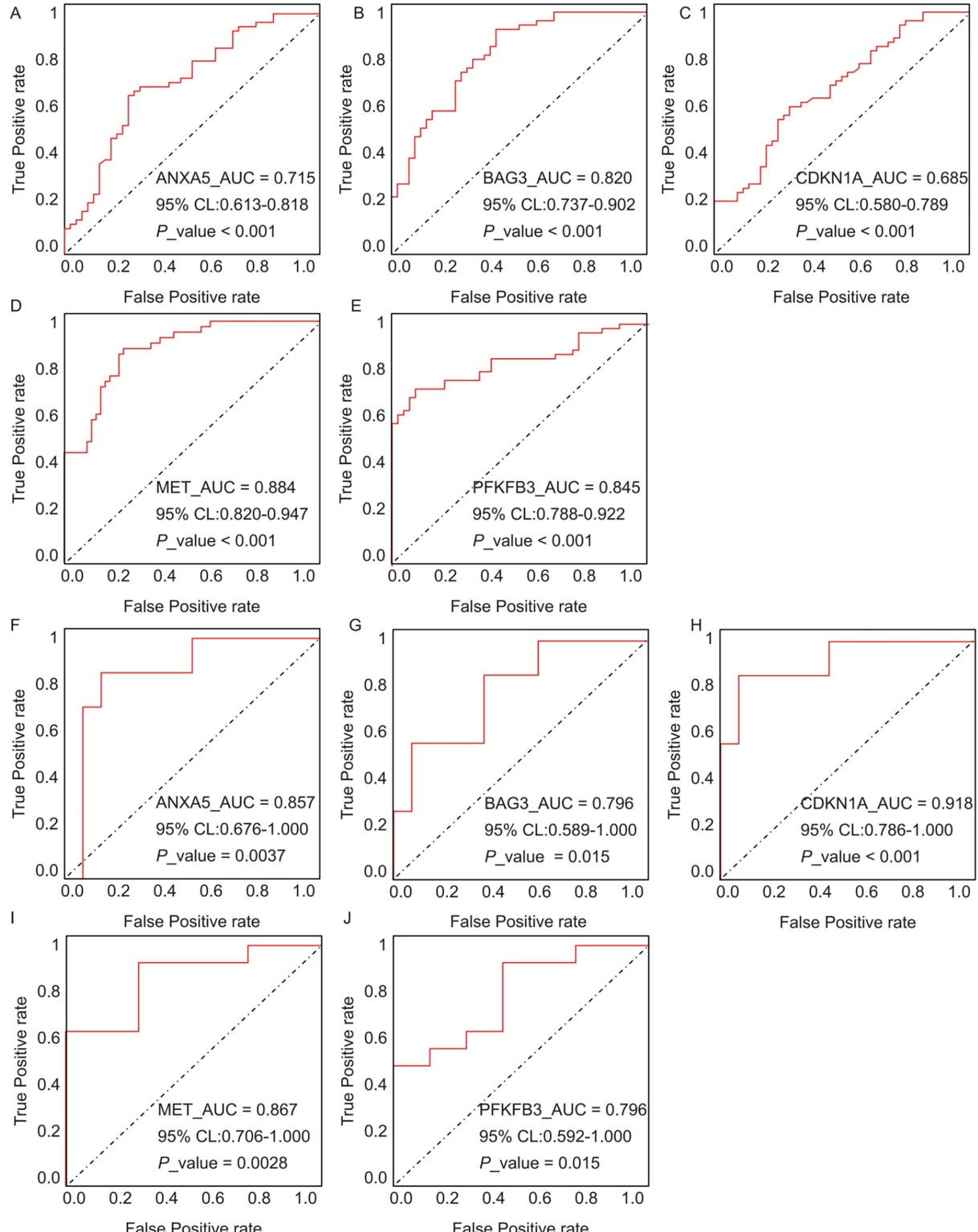

**Figure 10 ROC curves for the five hub genes in the validation datasets: (A–E) correspond to AD samples (GSE122063), and (F–J) correspond to T2DM samples (GSE64998), namely *ANXA5* for AD (A), *BAG3* for AD (B), *CDKN1A* for AD (C), *MET* for AD (D), *PFKFB3* for AD (E), *ANXA5* for T2DM (F), *BAG3* for T2DM (G), *CDKN1A* for T2DM (H), *MET* for T2DM (I), *PFKFB3* for T2DM (J).** The curves validate the diagnostic performance observed in training set, with annotations for AUC, 95% CI, and *P*-values.

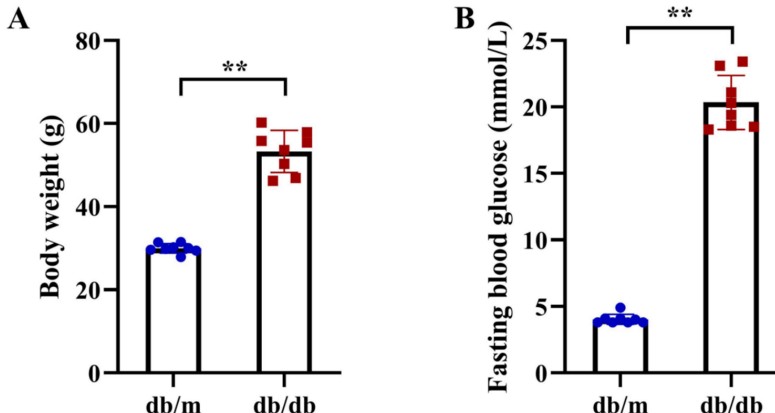

**Figure 11 Metabolic phenotypes of T2DM animal models.** (A) Body weight and (B) fasting blood glucose (FBG) levels in db/db mice (T2DM model, $n = 8$) and db/m control mice ($n = 8$) at 11 weeks of age (after 5 weeks of experimental feeding). Data were presented as mean ± SD. Statistical significance determined by independent Student's t-test. **$P < 0.01$.

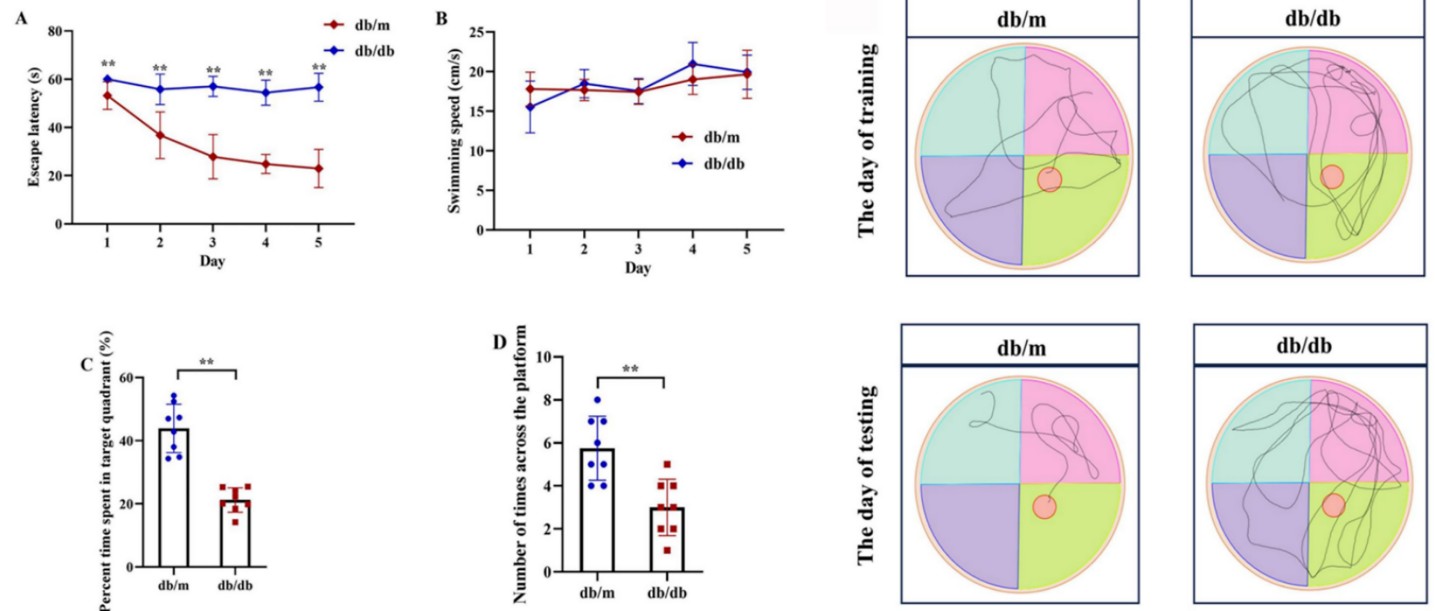

**Figure 12 Cognitive impairment in T2DM mice assessed by Morris Water Maze (MWM) test.** Comparison of Morris water maze performance between T2DM model (db/db mice, $n = 8$) and control group (db/m, $n = 8$), including escape latency in the 5-day orientation navigation test (A, compared by repeated-measures ANOVA, $P$ for group = 0.012), swimming speed in the 5-day orientation navigation test (B, compared by repeated-measures ANOVA, $P$ for group = 0.102), percent time spent in target quadrant in the spatial exploration test (C, compared by independent Student's t-test, $P < 0.001$), number of times across the platform in the spatial exploration test (D, compared by independent Student's t-test, $P < 0.001$) and representative swimming path in the spatial exploration test (E). Data presented as mean ± SD. **$P < 0.01$.

these four genes could serve as shared diagnostic biomarkers. In contrast, *PFKFB3* was significantly upregulated in AD patients but downregulated in T2DM patients, indicating that its expression may serve as a biomarker for diagnosing a single disease and may not be suitable for simultaneous diagnosis of both conditions. The area under the curve (AUC),

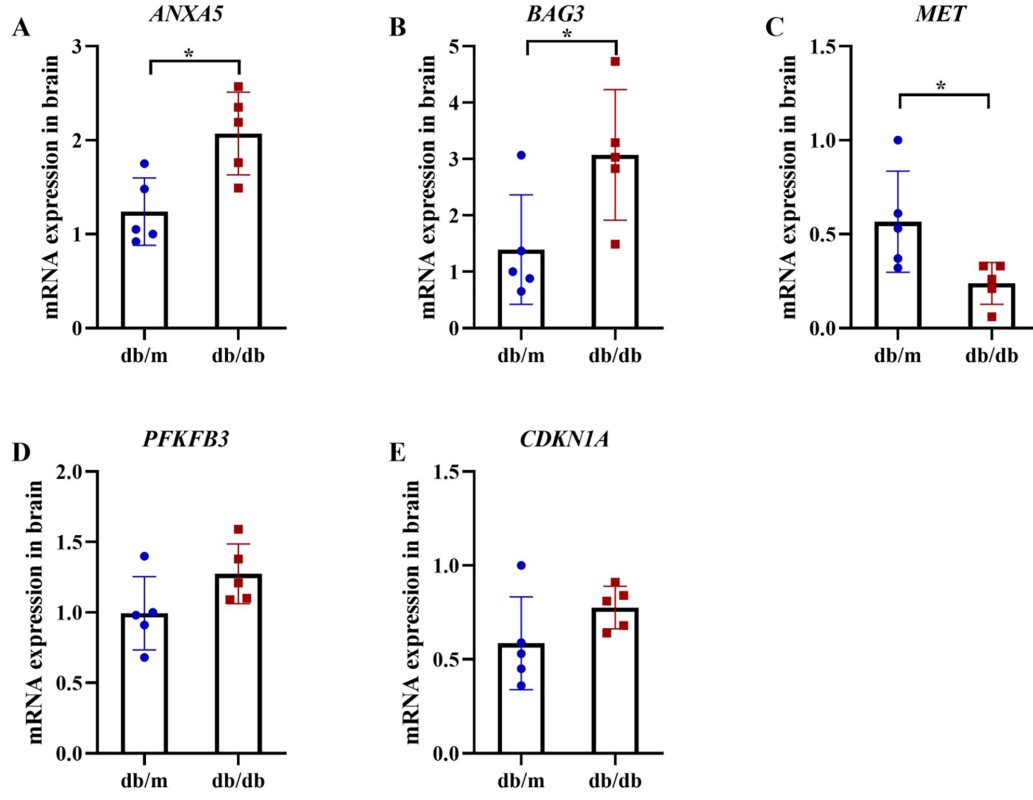

**Figure 13 qRT-PCR validation of hub genes in T2DM mouse brains.** qRT-PCR analysis of autophagy-related hub genes from Student's t-tests in brain tissues of db/db mice ($n = 5$) vs db/m controls ($n = 5$), including the expression of *ANXA5* (A, $P = 0.011$), *BAG3* (B, $P = 0.038$), *MET* (C, $P = 0.036$), *PFKFB3* (D, $P = 0.099$) and *CDKN1A* (E, $P = 0.156$). Data were presented as mean ± SD. Statistical significance determined by independent Student's t-test. *$P < 0.05$.

95% confidence intervals (CI), and *P*-values were calculated to quantify diagnostic accuracy. The results are visualized in Fig. 9 (training set) and Fig. 10 (validation set). In AD patients, the area under the curve (AUC) values for *ANXA5, BAG3, CDKN1A, MET,* and *PFKFB3* were 0.681, 0.698, 0.736, 0.762, and 0.794 in the training set (Figs. 9A–9E, all $P < 0.05$) and 0.715, 0.820, 0.685, 0.884 and 0.845 in validation set (Figs. 10A–10E, all $P < 0.05$). For T2DM patients, the AUC values for *ANXA5, BAG3, CDKN1A, MET,* and *PFKFB3* were 0.901, 0.913, 0.722, 0.87, and 0.866 in the training set (Figs. 9F–9J, all $P < 0.05$) and 0.857, 0.796, 0.918, 0.867, and 0.796 in validation set (Figs. 10F–10J, all $P < 0.05$). AUC values between 0.7 and 0.9 indicate good diagnostic accuracy, while values between 0.5 and 0.7 indicate moderate diagnostic accuracy. Based on this criterion, the five hub genes demonstrated excellent diagnostic performance for T2DM across both datasets ($0.9 > AUC > 0.7$). In AD, only *MET* and *PFKFB3* consistently showed good diagnostic performance, whereas the remaining genes exhibited moderate accuracy.

## Validation of candidate hub genes in animal models

Following the identification of 33 shared DEARGs and 12 hub genes, we applied a series of criteria to select the most robust candidate genes for further validation. The selection of

five hub genes (*ANXA5*, *BAG3*, *CDKN1A*, *MET*, *PFKFB3*) was based on their initial hub status (degree > 3 in the PPI network), consistent dysregulation across both training and validation datasets ($P < 0.05$ for Wilcoxon test) and sustained diagnostic utility (AUC > 0.6 in ROC analysis) in both AD and T2DM contexts across all datasets. We next verified the expression of the five identified hub genes in brain tissue samples from T2DM animal models (db/db mice) and corresponding controls (db/m mice) (Figs. 11A and 11B). Behavioral assessments using MWM revealed that learning and memory abilities were significantly impaired in the T2DM model compared to controls, as evidenced by reduced time spent in the target quadrant and fewer platform crossings during both training and testing days (Figs. 12A–12E). To further examine gene expression, we conducted qRT-PCR on brain tissues from both groups (Fig. 13). The results indicated that *ANXA5* and *BAG3* were significantly upregulated in db/db mice, whereas *MET* was significantly downregulated, relative to db/m controls (Figs. 13A–13C). In contrast, *CDKN1A* and *PFKFB3* expression did not differ significantly between the two groups (Figs. 13D and 13E). These findings partially diverge from bioinformatics results: *CDKN1A* was consistently upregulated in both AD and T2DM, while *PFKFB3* exhibited disease-specific expression patterns—upregulated in AD but downregulated in T2DM. These discrepancies may be attributed to interspecies differences in autophagic regulation or the limitations of monodisease models that do not account for tissue-specific pathophysiological contexts.

## DISCUSSION

AD and T2DM are two increasingly prevalent conditions that significantly impact public health and quality of life, posing substantial challenges to global health. This study highlights the shared pathophysiological mechanisms between the two diseases, particularly focusing on the role of autophagy, a critical cellular process responsible for maintaining homeostasis by degrading dysfunctional organelles and abnormal proteins. Dysregulated autophagy has been implicated in the progression of both AD and T2DM, suggesting that modulation of this pathway may offer therapeutic potential. Among these, *ANXA5* and *BAG3* were consistently upregulated, while *MET* was downregulated across both datasets and brain tissues from T2DM mouse models, supporting their potential roles as cross-disease biomarkers. These findings suggest a pathogenetic link between AD and T2DM mediated by key ARGs and are consistent with previous studies emphasizing the central role of autophagy in neurodegenerative and metabolic disorders.

The integrated analysis of DEARGs and WGCNA of AD/T2DM co-expression modules, combined with functional enrichment results, revealed that key ARGs and the autophagic pathway play critical roles in the pathophysiology of both diseases. Autophagy impairment, particularly the reduced capacity to degrade misfolded or aggregated proteins, is increasingly recognized as a central contributor to the pathogenesis of various diseases (*Kitada & Koya, 2021*), including AD (*Mancano et al., 2024*) and T2DM (*Arden et al., 2024*).

Our study identified 33 shared DEARGs between AD and T2DM patients, among which five hub genes, including *ANXA5, BAG3, CDKN1A, MET*, and *PFKFB3*, were validated. Intriguingly, disease-specific bioinformatics analyses revealed distinct clusters of hub DEARGs. For instance, the study focused on AD (*Li et al., 2023*) identified ten neurodegeneration-associated DEARGs, including five upregulated genes (*CAPNS1, GAPDH, LAMP1, LAMP2*, and *MAPK1*) and one downregulated gene (*CASP1*). In contrast, a T2DM-focused study (*Cui & Li, 2023*) revealed metabolism-related DEARGs, such as *GAPDH, LAMP2, FOXO3*, and *HSPA5*. Notably, the absence of overlap between disease-specific hub genes reported in these studies and the cross-disease candidates identified in our analysis may reflect differences in the datasets, discrepancies in bioinformatics methodologies, and distinct biological contexts. Cross-disease hub genes identified in this study may represent convergent targets for dual-pathology interventions, whereas the disease-specific DEARGs reported in other studies inform precision therapeutic strategies. Therefore, regulation of *ANXA5, BAG3, CDKN1A, MET*, and *PFKFB3* may represent a shared autophagy-related mechanism underlying AD-T2DM comorbidity, one that remains obscured in single-disease analysis.

Our study demonstrates that expression changes of *ANXA5, BAG3*, and *MET* in both AD and T2DM patients possess diagnostic potential (AUC > 0.6). *ANXA5* is a calcium-dependent phospholipid-binding protein with anti-inflammatory properties (*Li et al., 2022*). In models of atherosclerosis, ANXA5 has been shown to suppress inflammation by reducing the secretion of the pro-inflammatory cytokine TNF-α and increasing the anti-inflammatory cytokine IL-10 (*Burgmaier et al., 2014*). As a crucial regulator of autophagy, *BAG3* contributes to attenuating protein aggregation by promoting the clearance of pathogenic proteins such as tau in AD (*Carra, Seguin & Landry, 2008*). A recent study reported that *BAG3* upregulation in astrocytes is associated with decreased susceptibility to tau accumulation (*Sweeney et al., 2024*). *MET* encodes a receptor tyrosine kinase that regulates various cellular processes, including autophagy. While *MET* overexpression is implicated in tumor growth and metastasis in cancer (*Recondo et al., 2020*), its downregulation has been recently observed to be correlated with enhanced autophagy (*Seo et al., 2024*). The concurrent upregulation of *ANXA5* and *BAG3* alongside *MET* downregulation indicates significant dysregulation of autophagy in both AD and T2DM. It is possible that the upregulation of *ANXA5* and *BAG3* may reflect a compensatory response aimed at restoring autophagic activity to mitigate neuronal and metabolic deterioration. Conversely, *MET* downregulation may represent an adaptive mechanism that promotes autophagy to counter cellular stress arising from oxidative damage, inflammation, and metabolic dysfunction in AD and T2DM. Collectively, these three genes may serve as sensitive autophagy-related biomarkers involved in the clearance of toxic protein aggregates in both neuronal and metabolic tissues.

In contrast to the three consistently validated genes mentioned above, our integrated analysis revealed a regulatory divergence in *CDKN1A* and *PFKFB3*. While bioinformatics analysis identified *CDKN1A* upregulation in both AD and T2DM patients, experimental

validation did not confirm corresponding changes. The role of *CDKN1A* in promoting autophagy has been well documented in osteoarthritis (*Fang et al., 2024*) and hippocampal cells (*Althobaiti et al., 2024*). However, the role of *CDKN1A* in T2DM remains largely unexplored in the current literature. The lack of significant *CDKN1A* upregulation in brain tissues from T2DM animal models may be attributed to interspecies differences in autophagic response pathways. Unlike *CDKN1A*, *PFKFB3* exhibits paradoxical expression patterns—upregulated in brain tissues of AD but downregulated in adipose tissues of T2DM—reflecting its dual role as an autophagy modulator. For example, *PFKFB3* knockdown in podocytes activates autophagy and ameliorates diabetic nephropathy (*Zhu et al., 2021*), whereas the inhibition of *PFKFB3* can reduce autophagy in breast cancer cells (*La Belle et al., 2019*). Therefore, these inconsistencies may arise from tissue-specificity, species differences, and limitations inherent to experimental models. The opposing expression trends of *PFKFB3* suggest that its regulatory role in glycolysis and autophagy differs between neuronal and metabolic tissues. The db/db mouse model, while recapitulating key features of T2DM metabolic dysfunction, does not replicate AD-specific neuropathological hallmarks (*e.g.*, tau hyperphosphorylation/Aβ accumulation). This limitation may obscure the autophagy-related crosstalk and contribute to the observed divergence in *CDKN1A* function. Moreover, compensatory mechanisms in mice may mask gene expression changes relevant to human pathology. These findings underscore the necessity for validating candidate hub genes in human post-mortem tissues to ensure translational relevance.

While this study provides valuable insights into identifying hub ARGs as molecular bridges in AD-T2DM comorbidity, several limitations warrant further investigation. The use of distinct tissue types (brain for AD; adipose tissue for T2DM) introduces inherent tissue-specific biases in gene expression. Although WGCNA was performed independently for each tissue to reduce confounding effects, variability in cellular contexts may still influence the generalizability of shared hub genes. For example, *PFKFB3* exhibited opposing expression (upregulated in AD brain tissue but downregulated in T2DM adipose tissue), underscoring its tissue-specific regulatory roles.

Additionally, this study did not elucidate how the expression of the identified hub ARGs was influenced by environmental factors and lifestyle behaviors, neglecting epigenetic-environmental crosstalk that may regulate hub gene expression. AD, for instance, comprises sporadic late-onset AD (LOAD) and familial early-onset AD (EOAD), which are characterized by distinct genetic drivers (*e.g.*, *APOE* ε4 in LOAD *vs PSEN1/2* in EOAD) and may exhibit divergent patterns of autophagic dysregulation (*Hammers et al., 2025*). Similarly, T2DM is a heterogeneous condition, with subtypes driven by varying degrees of insulin resistance, β-cell dysfunction, or specific comorbidities such as cardiovascular disease and obesity (*Wang et al., 2024*). Furthermore, the influence of key demographic variables also requires careful consideration. Aging is the primary risk factor for both AD and T2DM, while different sex significantly impacts susceptibility to both diseases (*Moran et al., 2021*). Common comorbidities in T2DM patients (*e.g.*, hypertension, dyslipidemia, and chronic kidney disease) and environmental factors (*e.g.*, diet, physical activity, pollution) are also known to influence disease progression and could

modulate autophagy (*Ortega et al., 2024*). However, due to limited sample sizes in public datasets, we were unable to divide patients by disease subtype or account for comorbidities and lifestyle factors, thereby limiting our ability to distinguish their specific contributions to autophagy-mediated mechanisms in AD and T2DM. Future studies incorporating clinical phenotyping, lifestyle assessments, and molecular profiling are essential to elucidate these complex, multifactorial interactions.

Lastly, the current mouse models lack adequate validation. The use of distinct tissue types (brain tissue for AD and adipose tissue for T2DM) introduces inherent tissue-specific biases in gene expression. Although WGCNA was conducted independently for each tissue to reduce confounding effects, variability in cellular contexts may still limit the generalizability of the shared hub genes identified. While db/db mice serve as a well-established model for T2DM, characterized by hyperglycemia and insulin resistance, they do not fully recapitulate the neuropathological features of AD (*e.g.*, amyloid plaque formation and neurofibrillary tangles). This limitation necessitates future validation in human tissues exhibiting dual pathology—specifically, postmortem brain samples from patients with both AD and T2DM—to determine whether the identified hub genes represent shared pathogenic mechanisms within a unified biological context. In addition, comprehensive experimental validation utilizing rodent models with targeted manipulation (*e.g.*, gene knock-out or overexpression) of these hub genes will be crucial for understanding their shared pathogenic mechanisms.

## CONCLUSION

In summary, our multi-level approach, integrating bioinformatic analyses with experimental validation, has identified a set of hub ARGs shared between AD and T2DM, providing preliminary evidence for their potential roles in disease comorbidity. The consistent upregulation of *ANXA5* and *BAG3*, along with the downregulation of MET in both analyses, suggests that these genes may contribute to autophagy activation and help counteract the progression of neuronal and metabolic disorders. Notably, *CDKN1A* and *PFKFB3* exhibited divergent expression patterns, indicating complex regulatory mechanisms that may reflect tissue-specific autophagic responses to pathological changes. These findings underscore the need for further validation in larger clinical cohorts and disease-relevant models. Therefore, translating these findings into clinical applications requires efforts to integrate bioinformatics analyses with experimental and functional validation, ensuring that therapeutic strategies for AD and T2DM address both shared autophagy-related pathways and disease-specific pathobiology.

### Funding
This work was supported by the Youth Innovation Team Development Plan for Universities in Shandong Province (grant number: 2022KJ339). The funders had no role in study design, data collection and analysis, decision to publish, or preparation of the manuscript.

## Grant Disclosures

The following grant information was disclosed by the authors:
Youth Innovation Team Development Plan for Universities in Shandong Province:
2022KJ339.

## Competing Interests

The authors declare that they have no competing interests.

## Author Contributions

- Rui Zhang performed the experiments, analyzed the data, authored or reviewed drafts of the article, and approved the final draft.
- Ruowei Wang performed the experiments, analyzed the data, authored or reviewed drafts of the article, and approved the final draft.
- Shuna Zhai analyzed the data, prepared figures and/or tables, authored or reviewed drafts of the article, and approved the final draft.
- Chunhong Shen analyzed the data, prepared figures and/or tables, authored or reviewed drafts of the article, and approved the final draft.
- Yu An conceived and designed the experiments, authored or reviewed drafts of the article, and approved the final draft.
- Quanri Liu conceived and designed the experiments, authored or reviewed drafts of the article, and approved the final draft.

## Animal Ethics

The following information was supplied relating to ethical approvals (*i.e.*, approving body and any reference numbers):

All animal procedures were approved by the Animal Ethics Committee of Capital Medical University (AEEI-2023-296) and performed according to the relevant ethical standards.

## Data Availability

The raw data is available at GEO: GSE109887, GSE104674, GSE122063, and GSE64998.

The custom R scripts for bioinformatic analyses are available at GitHub:

- https://github.com/anyu900222/AD-T2DM-Autophagy,

- An, Y. (2025). Bioinformatic analysis of hub autophagy-related genes for type 2 diabetes mellitus and Alzheimer's disease. Zenodo. https://doi.org/10.5281/zenodo.16869519.

## Supplemental Information

Supplemental information for this article can be found online at http://dx.doi.org/10.7717/peerj.20143#supplemental-information.

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
