# Peer review of "Bioinformatic analysis and experimental validation of hub autophagy-related genes as novel biomarkers for type 2 diabetes mellitus and Alzheimer’s disease"

_PeerJ, doi:10.7717/peerj.20143_

## Round 0.1 · original submission · Major Revisions

I recommend the authors to address all the concerns raised by the reviewers.

·

Basic reporting

Title: Identification of Shared Autophagy-Related Hub Genes in Alzheimer’s Disease and Type 2 Diabetes Mellitus
This manuscript addresses a compelling and clinically relevant intersection between Alzheimer’s disease (AD) and Type 2 diabetes mellitus (T2DM) by investigating shared autophagy-related gene regulation. The authors present a combined bioinformatic and experimental strategy to identify common differentially expressed autophagy-related genes (DEARGs) and propose potential diagnostic markers and therapeutic targets.
Overstatement of Conclusions: While identifying shared DEARGs is valuable, several conclusions appear to overstate the data. In particular, the claims regarding the diagnostic and therapeutic potential of the identified hub genes—especially CDKN1A and PFKFB3—should be more cautiously framed. The inconsistencies between bioinformatics predictions and experimental validation for these genes suggest that further studies are needed before definitive claims can be made about their roles.

Experimental design

The manuscript would benefit from additional detail in the Methods section. Specific clarification aspects include the criteria for selecting DEARGs and defining hub genes, parameters used in WGCNA and enrichment analyses, statistical thresholds and correction methods (e.g., FDR), and details on animal model validation (e.g., number of animals, age, sex, experimental replicates).
The use of brain tissue for AD and adipose tissue for T2DM is biologically appropriate; however, it introduces a significant limitation for identifying shared molecular mechanisms due to inherent tissue-specific expression profiles. While WGCNA is a robust tool for uncovering gene co-expression modules, the current application does not address cross-tissue confounding. Standard normalization methods cannot eliminate expression variability arising from distinct cellular contexts. The authors should clarify whether WGCNA was performed separately or jointly for each tissue. If done jointly, the validity of the resulting modules is questionable. Integrative or tissue-aware network approaches should be considered, such as cross-tissue meta-analysis or adjustment for tissue-specific effects. This limitation should be explicitly acknowledged and discussed in the manuscript.

Validity of the findings

1. The manuscript acknowledges that CDKN1A and PFKFB3 exhibit discrepancies between in silico analysis and experimental findings. However, these genes still present a level of confidence not fully supported by the data. These discrepancies should be more explicitly discussed in the Results and Discussion sections, and the limitations in experimental validation should be emphasized.
2. The authors mention disease subtype variation and environmental influences as limitations, but these aspects warrant deeper discussion. Given the heterogeneity of AD and T2DM and the impact of aging, sex, and comorbidities on autophagy regulation, a more nuanced interpretation of the data is necessary.

Additional comments

1. The writing is generally clear but would benefit from light editing for grammar and clarity. Phrases like “we found that ME turquoise, ME blue…” are vague. More scientific phrasing would improve clarity. Grammar issues ("may may have a role...") and awkward phrasings ("declined the number of times that mice crossed..."). The manuscript repeatedly states that ANXA5, BAG3, and MET were validated across datasets and animal models. Streamlining this point could reduce redundancy (e.g., Lines 377–379 and again in Lines 405–420). Some sentences are overly complex or convoluted, which may impede reader comprehension. Line 423: " MET downregulation might facilitate autophagy as an adaptive response to cellular stress induced by oxidative stress, inflammation, and metabolic dysregulation in these diseases." This could be simplified and clarified.
2. “Autophagy” is incorrectly capitalized multiple times (e.g., Line 385). Line 410: “TNF-³” appears to be a formatting error; should be “TNF-α”. Some references are incomplete or misformatted (e.g., “Cui and Li(Cui & Li, 2023)” or “(Sweeney et al., 2024)” without full citations in this text). Make sure the reference list is appropriately formatted.
3. Consider updating or supplementing references with more recent literature in autophagy and comorbidity research.
4. The AUC analysis of diagnostic potential is detailed and includes training and validation cohorts. A log2FC threshold of 0.3 is relatively low and may lead to the inclusion of weak signals; more justification is needed for this choice.
5. Figure legends should be more descriptive to aid in the standalone interpretation of the results.
6. This manuscript offers a promising exploration of autophagy as a mechanistic link between AD and T2DM. The approach is innovative, and the preliminary findings are intriguing. However, the conclusions must be moderated, and methodological details should be expanded to allow reproducibility and enhance interpretability. With these revisions, the manuscript has the potential to make a meaningful contribution to the field.

·

Basic reporting

The manuscript is written in generally clear and professional English. Most sentences are grammatically correct and the scientific language is appropriate. However, there are occasional instances of awkward phrasing or dense sentence structure. For instance, lines 74–76 present a long, complex list of shared pathological features between AD and T2DM. Breaking these into two sentences would improve clarity.

The introduction provides a useful overview of the relevance of T2DM and AD comorbidity and introduces autophagy as a mechanistic link. However, the manuscript would benefit from more precise articulation of the knowledge gap. While Caberlotto et al. (2019) and Ye et al. (2023) are cited to support the connection between autophagy and AD/T2DM, it should be made more explicit how this study goes beyond these prior works. Adding more recent and mechanistically focused studies on autophagy's role in AD and T2DM would further strengthen the rationale.

Figures are relevant and mostly high-quality. However, figure legends could be more detailed. For example, Figures 2 A,B would benefit from clearer annotation and larger font axis labels. Meanwhile, abbreviations in figure captions (e.g., DEARGs, PPI) should be expanded upon first use.

The manuscript does not explicitly state where the raw data (e.g., qPCR values, Western blot quantifications) and code used for analysis are deposited. GEO datasets are mentioned, but accession numbers should be clearly listed in both Abstract and Methods. Considering provide a formal Data Availability Statement.

Experimental design

The research question is well defined and meaningful, aiming to address an underexplored intersection of two major diseases. However, more details are required for full reproducibility. For example:
1. Add database versions or access dates for ARGs.
2. Clearly describe the reason for filtering thresholds (e.g., FDR, log2FC)
3. Specify exact methods and thresholds used in hub gene ranking (e.g., degree, MCC).
4. Provide qPCR primer sequences or cite sources.

Validity of the findings

The identification of 33 shared DEARGs and 12 hub genes is clearly described. The integration of multiple datasets for validation is a major strength. ROC analysis is appropriately applied to assess diagnostic utility. However, the rationale for selecting the final five validated hub genes (ANXA5, BAG3, CDKN1A, MET, PFKFB3) should be explained more clearly (e.g., based on validation performance, biological plausibility, consistency across datasets).

Statistical methods are generally appropriate and follow accepted standards. However, several key details are missing or require clarification for full transparency and reproducibility: While multiple-testing correction is applied in differential expression analysis (e.g., FDR-adjusted p-values are noted), exact p-values are not consistently reported for qPCR, ROC, or behavioral analyses. These should be included wherever possible. The specific statistical tests used (e.g., t-test, ANOVA, Mann-Whitney U) are described in the Statistical Analysis section, but should also be clearly stated in figure legends or corresponding results for each comparison. For ROC curve analyses, 95% confidence intervals for AUC values are not provided. Including these would improve statistical rigor and enable better assessment of diagnostic precision. For qPCR validation, the normalization method is not described. Authors should specify which housekeeping gene was used and how normalization was performed.

Validation in both independent datasets and animal models improves robustness. Yet, the differences in tissue types (brain for AD, adipose for T2DM) should be acknowledged as a caveat. Highlighting the need for future validation in human tissues with dual pathology (e.g., autopsied diabetic AD patients) would be a valuable addition.

---

## Round 0.2 · accepted · Accept

The authors have satisfactorily addressed all the issues raised by the reviewers, so that the manuscript is acceptable in the present form.

·

Basic reporting

The manuscript has been significantly improved following the revision. I suggest acceptance

Experimental design

NA

Validity of the findings

valid

·

Basic reporting

The authors have addressed all previous concerns. Clarity, rationale, figures, and data availability are now adequately handled

Experimental design

The authors have thoroughly addressed reproducibility concerns. While hub gene prioritization could be expanded with multiple ranking metrics, the response is still satisfactory.

Validity of the findings

All previous concerns were thoroughly addressed. Transparency, robustness, and limitations are now clearly presented.